# Prevalence of Human Papillomavirus in the Oropharynx of Healthy Individuals in an Italian Population

**DOI:** 10.3390/jcm11071935

**Published:** 2022-03-31

**Authors:** Annalisa Palmieri, Dorina Lauritano, Agnese Pellati, Luca Scapoli, Claudio Arcuri, Luigi Baggi, Roberto Gatto, Francesco Carinci

**Affiliations:** 1Department of Experimental, Diagnostic and Specialty Medicine, University of Bologna, 40126 Bologna, Italy; annalisa.palmieri@unibo.it (A.P.); luca.scapoli2@unibo.it (L.S.); 2Department of Translational Medicine, University of Ferrara, 44121 Ferrara, Italy; agnese.pellati@unife.it (A.P.); crc@unife.it (F.C.); 3Department of Clinical Sciences and Translational Medicine, University of Rome “Tor Vergata”, 00113 Rome, Italy; claudio.arcuri@uniroma2.it (C.A.); luigi.baggi@uniroma2.it (L.B.); 4Department of Life, Health and Environmental Sciences, School of Dentistry, University of L’Aquila, 67100 L’Aquila, Italy; roberto.gatto@univaq.it

**Keywords:** human papilloma virus, oropharynx, oral cancer

## Abstract

Oral cavity and oropharynx cancer associated with human papil loma virus infection, particularly in young people who are continuously exposed to this virus, is a serious public health problem worldwide, especially for high-risk strains that are most associated with premalignant lesions and tumors. These neoplasms remain asymptomatic for a long time and, when they occur, they are already at an advanced stage. If diagnosed and treated early, oral cancer induced by human papilloma virus allows for high survival, as it often has a more favorable prognosis than oral cancers not directly related to viral infection. In this study, the presence of different high-risk and low-risk HPV strains was investigated to assess the epidemiological status in a population of healthy individuals. Two types of samples, one from the tonsils and one from the base of the tongue, were collected from 2015 healthy individuals of different sex and age. A total of 1885 DNA samples belonging to 1285 patients were tested for the presence of 11 high-risk HPV types plus 4 low-risk HPV types using real-time PCR. Of the patients’ DNA samples screened for 15 HPV types, only four samples were positive, all of which were taken from male smokers. These results indicate that newly acquired oral oncogenic HPV infections in the healthy population are rare and, in many cases, controversial. Therefore, more studies are needed to ensure fewer variations in outcomes and a greater clarification of HPV infection and its prevalence in the oropharynx of the healthy population, and to guide efforts to prevent the development of this infection which, if undiagnosed, can lead to the onset of malignant tumors in the oral cavity.

## 1. Introduction

The etiological role of human papillomavirus (HPV) infection in the development of squamous cell carcinoma of the uterine cervix has been extensively demonstrated [1,2]. In almost all cases, this tumor is caused by the persistence of high-risk viral strains, in particular HPV16 and HPV18 [3].

Due to the high tropism of the virus in the epithelium and similarities between genital and oropharyngeal epithelia, HPV is responsible not only for cervical tumors but also for 25% of head and neck squamous cell carcinoma (HNSCC) cases, particularly in the oropharyngeal region, with an incidence of HPV in up to 70% of cases [4]. The etiological role of tobacco and alcohol is well established, and micronutrient deficiencies and poor oral hygiene are also associated with a high risk of developing such neoplasms [5]. However, in recent decades an increasing prevalence of HPV in HNSCC has been reported, with a positivity rate up to 80% for this type of tumor [6].

To date, more than 200 HPV strains have been identified, which have been divided into two categories based on their neoplastic potential. Low-risk HPV genotypes, which include six types, 11, 40–44 and 54, are responsible for the development of benign lesions such as genital warts or respiratory papillomatosis. High-risk HPV genotypes, including types 16, 18, 26, 31, 33, 35, 39, 45, 51–53, 56, 58, 59, 66, 68, 70, 73, and 82, are related to the development of tumors in the oral cavity of the oropharynx, and in genitals and the anal area [7]. Following high-risk HPV infection, the expression of E6 and E7 proteins inhibits tumor suppressors RB and p53 and induces the malignant transformation of cells [5].

Several studies have evaluated the prevalence of human papillomavirus (HPV) in HNSCC, highlighting a wide variability of results linked to both the type of population and the location of the tumor, as well as the type of specimen and the method of virus detection [8,9,10,11,12,13].

HPV is more frequently found in tumors of the tonsillar crypts and the base of the tongue than in other head and neck districts, and HPV16 is reported as the predominant type detected [14,15,16].

Generally, HNSCC related to the presence of HPV arise at a relatively early age. Since the presence of HPV remains asymptomatic for a long time before generating a neoplasm, it is assumed that the infection occurs at a very early age [17].

Studies show that sexual contact is the most important risk factor, especially among young adults, and it has been shown that the risk increases linearly with an increasing number of sexual partners. Consequently, preventive considerations related to sex education could reduce the incidence of this cancer [18,19,20,21].

There has been increasing discussion on whether HPV-related HNSCC may be positively influenced by planned prevention and vaccination programs for cervical cancer-related HPV [22,23].

A secondary effect of cervical cancer screening and HPV vaccination could, therefore, be to reduce in parallel the number of cases of HNSCC related to the presence of HPV [7].

Although a direct relationship between the HPV vaccine and HNSCC prevention has not yet been demonstrated, there has been a significative reduction, up to 90%, of oropharyngeal HPV infections among vaccinated young adults in recent years [24,25].

This study aims to test the presence of multiple types of HPV (low and high risk) in a group of healthy male and female patients, over 18 years of age, in two specific anatomical areas, such as the base of the tongue and the tonsil, to investigate the prevalence of the virus in the oral cavity in a healthy population.

## 2. Materials and Methods

### 2.1. Patients’ Collection

A total of 2015 males and females (median age 30 years) who had given their consent were included in the study. Patients were recruited from the University of Tor Vergata (Italy) and from the University of L’Aquila (Italy). This study was approved by the L’Aquila ethical committee (approval number 26/2017).

The inclusion criteria were absence of systemic diseases (such as diabetes, heart failure, hypertension, renal failure, hepatic failure, respiratory failure, etc.), absence of facial trauma, good oral hygiene, and acceptance of the proposed follow-up protocols.

Patients who did not give their consent to conduct this study were excluded. Patients who could not guarantee adequate compliance, those with pathologies inducing systemic immunosuppression, patients who had undergone radio and/or chemotherapy, those with a positive history of taking immunomodulatory or immunosuppressive drugs, and patients with poor oral hygiene were also excluded.

Two samples were taken from each patient, one at the base of the tongue and one from the tonsil, using an oral swab that collects the desquamation epithelial cells. Swabs were immediately placed in a test tube containing silica gel capsules, which rapidly drIed the sample and preserved the integrity of the biological material.

The samples were stored at 4 °C until processing.

### 2.2. DNA Purification

Total DNA was extracted from each swab (tonsil and base tongue) with an automated procedure using a QIAcube HT extractor (Qiagen GmbH, Hilden, Germany) and the dedicated QIAamp 96 DNA QIAcube HT Kit (Qiagen GmbH, Hilden, Germany).

The samples were initially digested in a lysis buffer containing 20 mg/mL of proteinase K and then purified on silica membranes capable of simultaneously binding both genomic and viral DNA. The DNA was washed and then eluted in 150 µL of TE (10 mM Tris-HCl, 0.5 mM EDTA, pH 9.0).

Total DNA concentration and quality were measured using a NanoDrop 2000 spectrophotometer (Thermo Scientific, Waltham, MA, USA).

### 2.3. Primer Design

Each PCR oligonucleotide set was designed with the Primer-BLAST tool in order to identify primers that are specific to the intended HPV type (PMID: 22708584). This tool combines Primer3 software to design PCR primers and the BLAST and global alignment algorithms to screen primers against a user-selected database to avoid non-specific amplifications. For this investigation, the Papillomavirus Episteme (PaVE) database of curated human papillomavirus genomic sequences was used (accessed on 1 April 2019) [26].

Primer quality was further checked with MFEprimer v3.0 software, which also helped to set multiplex PCR assays [27]. This allowed us to assess primer compatibility and to verify whether specific amplification of targets was maintained when different sets of primers were combined.

Five multiplex real-time PCRs were performed for each sample. The primer and probe sequences used in the reactions are shown in Table 1. The first reaction detects and quantifies the amount of single human copies of the HMBS gene. The second and third reactions simultaneously detect and quantify the most common high-risk HPV types, as described by Moberg and colleagues [28], HPV 16, 31, 18, and 45, and HPV 33, 35, 39, and 58, respectively. The fourth reaction detects four other high-risk HPV types: 51, 56, and 66, while the fifth reaction quantifies low-risk HPV types: 44, 62, 72, and 84.

Human papillomaviruses 18 and 45 have been identified and quantified with a single probe marked with cyanine 5. Human papillomaviruses 33 and 58 have been identified and quantified with a single probe marked with FAM.

### 2.4. Real Time PCR

Absolute quantification assays were performed using ABI PRISM 7500 (Applied Biosystems, Foster City, CA, USA). Each reaction was performed in 20 µL containing 10 µL of 2X qPCRBIO Probe Mix Lo-ROX, 100 nM of DNA purified from samples, 200nM of each primer and 100 nM probe of each probe.

The amplification profile consists of an initial denaturation at 95 °C for 10 min, followed by a two-step amplification of 15″ at 95 °C and 60″ at 60 °C for 40 cycles. All PCR plates included non-template controls to ensure no contamination of reagents. Each experiment plate included non-template controls to ensure no contamination of reagents, and serial dilutions of the specific synthetic template. These positive controls were used to plot standard curves, i.e., threshold cycle values against the log of the copy number, which were used to check amplification efficiency and for quantification of targets in each sample.

The purified DNA obtained from each specimen was evaluated by real-time PCR using a unique human genome sequence as a target to quantify the number of copies analyzed in each reaction.

Standard curves for the human papillomavirus types were constructed in a multiplex reaction by using a mix of the same amount of different plasmids containing the target synthetic DNAs, in serial dilutions ranging from 10^1^ to 10^7^ copies. Serial dilutions of DNA extracted from blood, with amounts ranging from 10^1^ to 10^5^ copies, were used as the standard to quantify the human genome copy number in each reaction.

### 2.5. Statistical Analysis

Descriptive statistics and data analysis was performed using SPSS software v.25 (IBM, New York, NY, USA).

## 3. Results

A total of 2015 volunteers who signed an informed consent form were recruited for this study. The clinicians performed two oral swabs, the first at the base of the tongue and the second at the tonsil. The specimens were sent to the laboratory by express courier. The laboratory received and processed 1882 specimens from the base of the tongue and 1844 from the tonsil. Missing specimens were due to a number of reasons, including patients refusing to give consent for sampling at the chair, clinician decisions, inadequate conservation of samples, or lost shipment. About half of all samples did not pass the minimal amount threshold of 1000 human cell genomes; this threshold was set to reduce the chance of false negatives of the HPV assay due to the low DNA concentration. The samples over the threshold were 971 from the base of the tongue and 914 from the tonsil, with median amounts of DNA of 6573 (IQR: 3946–11148) and 4381 (IQR: 2240–8652) human genome equivalents, respectively. These samples were then tested for the presence of 11 high-risk HPV types plus 4 low-risk HPV types. The screening involved 1285 patients, of which 600 were tested with two specimens from base of the tongue and tonsil, 371 were tested with a specimen from the base of the tongue only, and 314 were tested with a specimen from the tonsil only. The pyramid plot shows the age and gender distributions of patients (Figure 1) and reveals an overrepresentation of the age class of between 20 and 30 years and an overrepresentation of female patients. Additional patient data are reported in Table 2.

Among the 1885 DNA samples from 1285 patients that were screened for 15 types of HPV, only four samples returned positive results (Table 3). All of these were collected from the base of the tongue and represented patients identified as middle-aged, male smokers. Only one tonsil sample was available from these four individuals, and this tested negative for HPV.

The calculated prevalence of any of the tested HPV types (4 out of 1285 patients) was 0.31% (95% C.I. 0.12–0.80), and prevalence for high-risk HPV (3 out of 1285 patients) was 0.23% (95% C.I. 0.08–0.68). The prevalence values among the 971 samples from the tongue were 0.42% (95% C.I. 0.16–1.05) and 0.31% (I.C. 95%: 0.11–0. 90) for any HPV and high-risk HPV, respectively.

The vaccination status of the participants was not recorded in the study. In the attempt to evaluate the putative number of participants completely vaccinated against HPV, we referred to data published at the Italian Ministry of the Health web site (https://www.salute.gov.it/portale/documentazione/p6_2_8_3_1.jsp?lingua=italiano&id=27 (accessed on 1 December 2021), which reports the percentage of the population who are fully vaccinated, according to age, date of birth, and gender in the area of recruitment (Table 4). On this basis, we calculated that in our sample the number of fully vaccinated patients is around 51 out of 1285.

## 4. Discussion

The alarming increase in HPV-associated cancer of the oral cavity and oropharynx, and the simultaneous prevalence of human papillomavirus infection, particularly in young people who are continuously exposed to this virus, is becoming a major public health problem worldwide [29,30].

The two high-risk strains most commonly associated with pre-malignant lesions and tumors are HPV-16 and HPV-18. In fact, over 70% of OPSCC cases and over 50% of tonsillar cancers in the United States (USA) have been associated with high-risk HPV serotypes [31].

The high spread of this virus has meant that HPV-associated OPSCC is now the most common cancer in the United States, surpassing cervical cancer [32].

The neoplasm caused by HPV remains asymptomatic for a long time and when it manifests it is already in an advanced stage. If diagnosed and treated early, HPV-induced cancer allows for 85–90% survival, and HPV-positive OPSCCs have been shown to have a better prognosis than HPV-negative ones [33,34].

HPV infection occurs mainly by sexual transmission, and it is estimated that over 60% of adults have contracted the virus at least once in their lifetime [20,35,36]. Fortunately, in immunocompetent subjects, the infection has a rapid course which is often asymptomatic, and the virus is destroyed without further consequences.

Only in 10% of cases, especially in cases of HPV 16 and 18 infections, does it turn into persistent infection and cause precancerous lesions. Precancerous lesions in turn usually take a few years to develop into infiltrating carcinoma [37,38].

Since oral and pharyngeal cancer is diagnosed in 70% of cases at an advanced stage, surgery is the only therapeutic tool that can be used with a consequent large waste of economic resources [39,40].

For these reasons, it is important to have an overview of the epidemiological situation in the healthy population. Although members of this population may be asymptomatic, they could be HPV positive with the potential to develop a precancerous lesion over many years.

In this study, the presence of different high-risk and low-risk HPV strains was examined to assess the epidemiological situation in a study sample consisting of healthy individuals of different age groups.

The screening involved 1285 patients, of which 600 were tested with two specimens from the base of the tongue and tonsil, 371 tested with a specimen from the base of the tongue only, and 314 tested with a specimen from the tonsil only.

The gender distribution of age class was between 20 and 30 years and the majority of participants were female.

These samples were tested for the presence of 11 high-risk HPV types plus 4 low-risk HPV types.

Among the 1885 DNA samples from 1285 patients that were screened for 15 types of HPV, only four samples returned positive results. All of them were collected from the base of the tongue of middle-aged, male smokers. Only one tonsil sample was available from these four individuals, this tested negative for HPV.

Low rates of infection in tonsil tissue are consistent with higher immunological exposure in the epithelium of the tonsils than in cervical epithelia. This site contains IgA and proteolytic enzymes that protect against HPV infection [41].

It was also found that, in the young population, most HPV infections are effectively eliminated by the immune system within 6–12 months [42,43].

There are numerous conflicting results described in the literature on the presence of HPV in the oral cavity and its correlation with any infections at the genital level or with sexual habits. In fact, the prevalence percentage varies from 0% to 50% [9,10,12,13].

This high heterogeneity in HPV prevalence across countries and cultures could be due to a number of factors that may be intrinsic to the type of study performed, such as differences in sample collection, processing, and testing, as well as the small sample size which represents a limitation for the correct assessment of prevalence [44].

Kreimer et al. [45], in a study evaluating the prevalence of HPV in healthy adult men from three countries, the United States, Mexico, and Brazil, found that HPV16 infection was rare in healthy men, especially at a young age, and was positively associated with current tobacco use. Furthermore, oral HPV infection was not associated with oral sexual behaviors.

The same authors, in a systematic review of the literature, showed that the prevalence of oral HPV16 was 1.3% among healthy individuals and appeared to differ according to the geographic region of origin [44].

Castro et al. [46] conducted a pilot study in order to verify the presence of HPV DNA in the oral and genital mucosa of patients with genital HPV infection using a PCR technique. The results show that the percentage of HPV infection was higher in the genital area (57%) compared to oral mucosa (0%), suggesting that genital HPV is not a predisposing factor for oral HPV infection in the same patient.

Another study reported that oral HPV16 infection is present in only 1% of cancer-free individuals [47].

A bimodal age distribution in HPV infection has been described, with the prevalence peaking at 30–34 and 60–64 years. Some studies suggest that the higher prevalence of oral HPV in old age could be caused by a longer duration of infections, rather than a higher incidence [47,48].

## 5. Conclusions

Our study, based on a large sample of healthy young individuals, demonstrates that the oral cavity is not a reservoir of HPV in healthy people. Therefore, it is worth understanding the vaccination status of study participants to assess what roles HPV vaccines may have in the protection against oral HPV infections and, ultimately, against OPSCC-HPV.

More studies are needed which have fewer variables on outcomes and greater clarification of HPV infection and its prevalence in the oropharynx of the healthy population, to guide efforts to prevent the development of this infection which, if undiagnosed, can lead to the onset of malignant neoplasms of the oral cavity.

Oral healthcare professionals are in an excellent position to screen for and detect oropharynx cancers thorough intra- and extra-oral examinations, which should be emphasized for the early detection of these types of cancers.

## Figures and Tables

**Figure 1 jcm-11-01935-f001:**
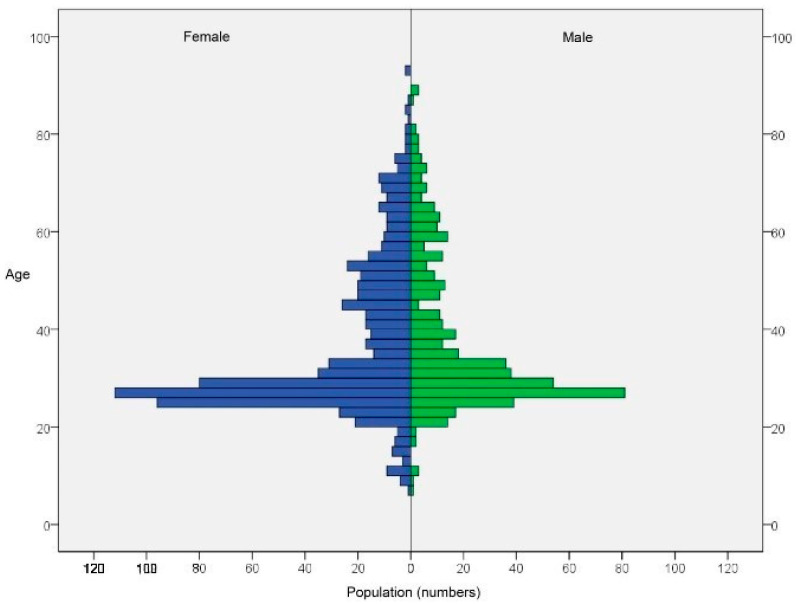
Age and gender distributions of patients.

**Table 1 jcm-11-01935-t001:** Primer and probe sequences.

**First reaction: quantification of human genome**		
**Gene symbol**	**Primer sequences 5′-3′**	**Probe sequences 5′-3′**
HMBSA	f-AAGACACGTTCCACTTTTGATTCA	AAGCCTCCGAACTGCACACAAACGTC-JOE
	r-ACACAAAAAGAAGGCGCACTTC	
**Second reaction: detection of HPV 16, 18, 31, and 45**		
**HPV type**	**Primer sequences 5′-3′**	**Probe sequences 5′-3′**
HPV16	f-AGCTCAGAGGAGGAGGATGAA	CCAGCTGGACAAGCAGAACCGG-FAM
	r-GGTTACAATATTGTAATGGGCTC	
HPV18	f-CATTTTGTGAACAGGCAGAGC	AGAGACAGCACAGGCATTGTTCCATG -JOE
	r-ACTTGTGCATCATTGTGGACC	
HPV31	f-ACGATTCCACAACATAGGAGGA	CTCCAACATGCTATGCAACGTCC-CY5
	r-TACACTTGGGTTTCAGTACGAGGT	
HPV45	f-CATTTTGTGAACAGGCAGAGC	AGAGACAGCACAGGCATTGTTCCATG -JOE
	r-CAACACCTGTGCATCATTCTGA	
**Third reaction: detection of HPV 33, 35, 39, and 58**		
**HPV type**	**Primer sequences 5′-3′**	**Probe sequences 5′-3′**
HPV33	f- CGTCGCAGGCGTAAACG	AGATGTCCGTGTGGCGGCCTAG-FAM
	r-ACAGGAGGCAGGTACAC	
HPV35	f-ACCCATACCAAAGCCTGCTC	ACGACTTCGAGGGGGTACCGAGCTCCCC-JOE
	r-GCACTGAGTCGCACTCGC	
HPV39	f-CGAGCAATTAGGAGAGTCAGAGG	AACCCGACCATGCAGTTAATCACCAAC-CY5
	r-TGTGTGACGCTGTGGTTCAT	
HPV58	f-GCGTCGCAGACGTAAACG	AGATGTCCGTGTGGCGGCCTAG-FAM
	r-ACAGGAGGCAGGTACAC	
**Fourth reaction: detection of HPV 51, 56, and 66**		
**HPV type**	**Primer sequences 5′-3′**	**Probe sequences 5′-3′**
HPV51	f-CACCGCCTCCACCTTTGT	GGCGCCCAAGACGCCGCGGTATCCC-CY5
	r-GTGTGCGTAGGACTCTCTGG	
HPV56	f-GCTAACCTACTGGAGGACTGG	CCCCGCCAGTGGCCACCAGCCTAGA-JOE
	r-TTCTGTTGGTGGCTGTTCCC	
HPV66	f-CACCGCCTCCACCTTTGT	GGCGCCCAAGACGCCGCGGTATCCC-FAM
	r-GTGTGCGTAGGACTCTCTGG	
**Fifth reaction: detection of HPV 44, 62, 72, and 84**		
**HPV type**	**Primer sequences 5′-3′**	**Probe sequences 5′-3′**
HPV44	f-TGCTTCACACTCCTCCTCCT	GCACCGCCGAGGACTGCGTGGACGC-CY3
	r-CCTCGGGGTCGTTTACATGG	
HPV62	f-CTGGACGACCTGCACCTAAC	GACCTGTCCGCCGGTGAACTGCTGTCCT-CY5
	r-ACGTGTAGCTCCCGTATTGC	
HPV72	f-TCTGCAACGGACCTGTATCG	TGCAAACAGGCGGGTACCTGCCCTCCTG- FAM
	r-AACTGGCCCACTTCAGGAAC	
HPV84	f-CAGCCCGACTCTACACAAGG	ACACCGGCCGGTTGACAGTTGCAGCAC-JOE
	r-AGTTACTGTGTTCCCGTGGC	

**Table 2 jcm-11-01935-t002:** Descriptive data of participants screened for HPV high-risk strains.

	No		Yes		Missing Data
**Male**	748	60%	498	40%	39
**Smoking**	801	64%	446	36%	39
**Alcohol**	781	63%	466	37%	39
**Partner**	473	38%	773	62%	39

**Table 3 jcm-11-01935-t003:** Details of positive samples.

Individual	Tongue Base	Tonsil	Sex	Age	Smoke	Alcohol	Partner
FC95	HPV45: 3385 copies; 0.76 HPV/cell	no sample	male	49	yes	yes	no
FC228	HPV16: 3268 copies; 0.75 HPV/cell	no sample	male	55	yes	yes	no
FC151	HPV18: 263 copies; 0.03 HPV/cell	no sample	male	60	yes	no	no
FC206	HPV44: 1821 copies; 0.78 HPV/cell	negative	male	62	yes	yes	yes

**Table 4 jcm-11-01935-t004:** Percentage of population fully vaccinated for high-risk HPV stratified by age and gender.

Date of Birth	Age	Female	Male
1995	24	21%	0%
1996	23	48%	0%
1997	22	65%	1%
1998	21	67%	1%
1999	20	69%	1%
2000	19	70%	2%
2001	18	70%	3%
2002	17	65%	4%
2003	16	71%	3%
2004	15	67%	3%
2005	14	60%	6%
2006	13	56%	30%
2007	12	29%	15%

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
