# Peer review of "Prevalence of Human Papillomavirus in the Oropharynx of Healthy Individuals in an Italian Population"

_jcm, 2022, doi:10.3390/jcm11071935_

Round 1

Reviewer 1 Report

Review for Manuscript ID:  Manuscript ID: jcm-1655899 entitled "Prevalence of human papillomavirus in the oropharynx of healthy individuals in an Italian population”

There are points need to be corrected as follow: 

  • In the abstract, the prevalence of positive cases according to gender need to be mentioned.
  • In general, there are some extra lines which do not need to be as separate paragraph. Therefore, a proofread is necessary.
  • In introduction, paragraph 3 needs to be part of paragraph 2.
  • Last line of introduction “to better understand the pathogenesis 80 and future clinical therapeutic strategies” need to be removed as this is not part of the aim of the study.
  • Statistical analysis need to be added to the end of materials and methods.
  • Result: number of patients in result and materials and methods are 2015, which is not consistent with the abstract (1285 subjects). These number are confusing.
  • Result: The line “The purified DNA obtained from each specimen was evaluated by real-time PCR using a unique human genome sequence as a target to quantify the amount of copies analyzed in each reaction” need to be removed as it is part of materials and methods.
  • Discussion: Explanation of variation of HPV prevalence between different culture and countries need to be explained.
  • Add limitation of the study.

Regards, 

Author Response

Ferrara 24/03/2022

            Dear Editor,

attached please find the revised version of our original manuscript entitled " "Prevalence of human papillomavirus in the oropharynx of healthy individuals in an Italian population” (Manuscript ID: jcm-1655899).

All concerns raised by the reviewer were addressed in the text (highlighted in green) and in this letter.

Best regards

Answers to reviewer 1

  • In the abstract, the prevalence of positive cases according to gender has been mentioned: “…all of them were smokers’ males”.
  • Extra lines have been removed and proofreading has been done.
  • In the introduction, paragraph 3 has been attached to paragraph 2.
  • The last line of the introduction “to better understand the pathogenesis and future clinical therapeutic strategies” has been removed.
  • The statistical analysis section has been added to the end of materials and methods.
  • The number of patients in the abstract has been detailed: “Two types of samples, one from the tonsil and one from the base of the tongue, were collected from 2015 healthy individuals of different sex and age. 1885 DNA samples belonging to 1285 patients were tested for the presence of 11 high-risk HPV types plus 4 low-risk HPV types by Real-time PCR”.
  • The line “The purified DNA obtained from each specimen was evaluated by real-time PCR using a unique human genome sequence as a target to quantify the number of copies analyzed in each reaction” has been removed from “Results” and moved in “Materials and methods section”.
  • Variation of HPV prevalence between different culture and countries have been better explained in the “Discussion” section. “This high heterogeneity in HPV prevalence across countries and cultures could be due to a number of factors that may be intrinsic to the type of study performed, such as differences in sample collection, processing, and testing, as well as the small sample size that it represents a limitation to the correct assessment of prevalence (44)”.
  • Limitations of the study have been added in the “Conclusion” section: “Our study based on a large sample of healthy young individuals demonstrates that oral cavity is not a reservoir of HPV in healthy people, therefore should be interesting to understand the vaccination status of the participants in the study to assess what roles HPV vaccines may have to protect against oral HPV infections and ultimately against OPSCC-HPV related”.

Yours sincerely,

Prof. Dorina Lauritano

Reviewer 2 Report

Line 55: you mentioned several studies. Please cite which studies are you referring to.

Line 65: you mentioned several studies. Please cite which studies are you referring to.

Line 84: mention the IRB number approval from your hospital/university.

Line 124: It is great that you insert those tables. Please edit them in a way it will be easier for the readers to follow the data from them.

Line 151: where is your statistical analysis text? I cannot find it in M&M chapter. Please insert.

Results: Please insert statistical analysis.

Discussion: In the discussion chapter you should write how HPV may determine other types of cancer. It is well known that is associated with other form not only fo H&N region. Write about how your results will help dental practitioners specialised in treating those types of patients. what could be the advantages of using your assessment. 

Write about the strenghts and limitations of your study. Write wha should be done in future studies.

Conclusion: I cannot find your conclusion chapter.

Author Response

Ferrara 24/03/2022

            Dear Editor,

attached please find the revised version of our original manuscript entitled " "Prevalence of human papillomavirus in the oropharynx of healthy individuals in an Italian population” (Manuscript ID: jcm-1655899).

All concerns raised by the reviewer were addressed in the text (highlighted in green) and in this letter.

Answers to reviewer 2

  • References have been added to the sentence at line 55: “Several studies have evaluated the prevalence of human papillomavirus (HPV) in HNSCC, however, highlighting a wide variability of the results linked to both the type of population and the location of the tumor, as well as the type of specimen and the method of virus detection (8-13)”.
  • References have been added to the sentence at line 65: “Studies show that sexual contact is the most important risk factor, especially among young adults, and it has been shown that the risk increases linearly with an increasing number of sexual partners. Consequently, preventive considerations related to sex education could reduce the incidence of this cancer (18-21)”.
  • The IRB number approval has been mentioned in the “Materials and methods” section: “The study was approved by the L'Aquila ethical committee (approval number 26/2017)”
  • Tables have been edited
  • “Statistical analysis” section has been added at the end of the “Materials and methods” section
  • The mechanism of how HPV may determine other types of cancer have been added in the “Introduction” section: “High-risk HPV genotypes including types 16, 18, 26, 31, 33, 35, 39, 45, 51–53, 56, 58, 59, 66, 68, 70, 73, and 82 are related to the development of tumors in the oral cavity of the oropharynx and genitals including anal areas (7). Following high-risk HPV infection, the expression of E6 and E7 proteins inhibits tumor suppressors RB and p53 and induce cells malignant transformation (5)”.

  • How our results will help dental practitioners specialized in treating those types of patients, have been explained in the “Conclusion” section: “Oral healthcare professionals are in an excellent position to screen for and detect oropharynx cancers thorough intra- and extra-oral examination that should be over-emphasized for early detection of these types of cancers.”
  • Strengths and limitations of our study and future prospective have been added in the “Conclusion” section: “Our study based on a large sample of healthy young individuals demonstrates that oral cavity is not a reservoir of HPV in healthy people, therefore should be interesting to understand the vaccination status of the participants in the study to assess what roles HPV vaccines may have to protect against oral HPV infections and ultimately against OPSCC-HPV related. More studies are needed to have fewer variables on outcomes and more clarification on HPV infection and its prevalence in the oropharynx of the healthy population, to guide efforts to prevent the development of this infection which, if undiagnosed, can lead to the onset of malignant neoplasms of the oral cavity.”
  • “Conclusion” section has been added

Yours sincerely,

Prof. Dorina Lauritano

Round 2

Reviewer 2 Report

Thank you for answering my question. Now, the paper is in good terms.

I wish you best of luck with many great researches.

Best regards